# Separation of Dihydro-Isocoumarins and Dihydro-Stilbenoids from *Hydrangea macrophylla* ssp. *serrata* by Use of Counter-Current Chromatography

**DOI:** 10.3390/molecules27113424

**Published:** 2022-05-26

**Authors:** Johannes Wellmann, Beate Hartmann, Esther-Corinna Schwarze, Silke Hillebrand, Stephan I. Brueckner, Jakob Ley, Gerold Jerz, Peter Winterhalter

**Affiliations:** 1Institute of Food Chemistry, Technische Universität Braunschweig, Schleinitzstraße 20, 38106 Braunschweig, Germany; g.jerz@tu-braunschweig.de (G.J.); p.winterhalter@tu-braunschweig.de (P.W.); 2Symrise AG, Mühlenfeldstraße 1, 37603 Holzminden, Germany; beate.hartmann@symrise.com (B.H.); esther-corinna.schwarze@symrise.com (E.-C.S.); silke.hillebrand@symrise.com (S.H.); stephan.brueckner@symrise.com (S.I.B.); jakob.ley@symrise.com (J.L.)

**Keywords:** *Hydrangea macrophylla* ssp. *serrata*, dihydro-isocoumarins, dihydro-stilbenoids, cyanogenic glycosides, flavonols, secoriridoids, counter-current chromatography, liquid chromatography, ion-mobility high-resolution mass spectrometry, nuclear magnetic resonance spectroscopy

## Abstract

Previously, different *Hydrangea macrophylla* ssp. *serrata* cultivars were investigated by untargeted LC-MS analysis. From this, a list of tentatively identified and unknown compounds that differ significantly between these cultivars was obtained. Due to the lack of reference compounds, especially for dihydro-isocoumarins, we aimed to isolate and structurally characterise these compounds from the cultivar ‘Yae-no-amacha’ using NMR and LC-MS methods. For purification and isolation, counter-current chromatography was used in combination with reversed-phase preparative HPLC as an orthogonal and enhanced purification workflow. Thirteen dihydro-isocoumarins in combination with other metabolites could be isolated and structurally identified. Particularly interesting was the clarification of dihydrostilbenoid glycosides, which were described for the first time in *H. macrophylla* ssp. *serrata*. These results will help us in further studies on the biological interpretation of our data.

## 1. Introduction

*Hydrangeaceae* is an economically important plant genus in the upper segment of commercialized ornamentals. Among wild plants of *Hydrangea macrophylla* ssp. *serrata* in Japan there are plants whose leaves contain sweet tasting phyllodulcin, a phenyl-dihydro-isocoumarin [1]. In its native Japan, the dried leaves of the variety *H. macrophylla* ssp. *serrata* var. *thunbergii* are known as a tea, or sweetener for tea, and consumed during the Japanese flower festival, ‘Hana matsuri’. While plants that accumulate phyllodulcin were cultivated in various parts of Japan, they are originally distributed in the wild in natural populations [2].

So far, the constitution of metabolites from different *Hydrangeaceae* were extensively investigated by preparative studies. In addition to dihydro-isocoumarin, flavonol, cyanogenic glycoside, coumarin, secoiridoid, and cinnamic acid derivatives were described [3,4,5,6,7,8,9,10,11,12,13,14,15,16,17,18,19,20,21,22,23,24,25,26,27,28,29,30,31,32,33,34,35,36,37]. In a previous study, we examined different *H. macrophylla* ssp. *serrata* cultivars through comprehensive metabolite profiling via UPLC-ESI-IMS-QToF analysis. The untargeted screening of different *Hydrangea macrophylla* ssp. *serrata* cultivars resulted in a list of metabolites that differed significantly between the investigated plants (data not shown here). Some of the selected metabolites were tentatively identified through literature search. The quantitatively dominant metabolites are dihydro-isocoumarin glucosides, especially hydrangenol-8-O-glc, thunberginol G-3′-O-glc and phyllodulcin-8-O-glc which were already known from previous studies. Although over 100 substances were described previously, a relatively small number of substances can be tentatively assigned. Therefore, we decided to isolate the unknown compounds in order to simplify the biological interpretation of the collected data.

A well-suited tool for the isolation of natural products is counter-current chromatography (CCC). This technique is based on different liquid–liquid partitioning processes of solutes in two immiscible phase layers of a solvent system. Modern CCC instruments consists of two or three self-balanced bobbins with multilayer coiled columns (mostly PTFE tubings), which are rotated in a planetary motion. Due to the centrifugal force fields, a multitude of partition processes occur within the column-coil tubing systems, which cause the separation of substances with different partition coefficients (*K_D_*) in the used solvent system [38,39]. Main advantage of this technique is the separation of several grams of sample material in a relatively short time without significant sample loss due to neglectable chemisorption [40,41]. Various devices (high-speed CCC, high-performance CCC, and centrifugal partition chromatography CPC) were developed over the past decades, that essentially rely on the same separation principles. High-performance CCC (HPCCC) represents a further development compared to high-speed CCC (HSCCC), with higher centrifugal forces and, therefore, enhanced stationary phase retention, resulting in improved chromatographic resolution [42]. Modern HSCCC and HPCCC methods are widely used in combination with structure elucidations tools, such as NMR and LC-MS, for separation and identification of natural products [43,44,45,46,47,48]. However, methods for the purification and isolation of metabolites from extracts of *H. macrophylla* ssp. *serrata* so far were not described, particularly viewing on the recovery of dihydro-isocoumarins.

The aim of this work was to identify further metabolites, especially those ones that were investigated in the untargeted screening of different *Hydrangea* cultivars (data not shown here). Therefore, the fractionation and isolation of metabolites from a methanolic extract of *H. macrophylla* ssp. *serrata* using HPCCC and HSCCC techniques is presented. To the best of our knowledge, this is the first description using countercurrent chromatography for separation and purification of secondary natural products from *Hydrangea macrophylla* ssp. *serrata* extract. Since neither commercially available reference substances nor analytical data, in particular mass spectrometric MS/MS data, are available for dihydro-isocoumarins, comprehensive mass spectrometric and spectroscopic data for the isolated substances using ESI-IMS-QToF and NMR measurements are provided, which can be used in further studies.

## 2. Results and Discussion

Sample material from a *H. macrophylla* ssp. *serrata* cultivar named ‘Yae-no-amacha’ was selected based on the results from untargeted screening, due to its diverse composition of interesting compounds from the group of dihydro-isocoumarins (data not shown here). Therefore, plants were propagated via cuttings and then cultivated in experimental fields (Holzminden, Lower Saxony, Germany). In total, 1.2 kg of dried leaves were harvested by pruning of the plants in late August 2021. Subsequently, the dried leaves were macerated with methanol and the crude extract was filtered and concentrated by vacuum evaporation. The resulting extract was then suspended in bi-distilled water, partitioned successively with *n*-hexane and ethyl acetate. After concentration by vacuum evaporation, three different fractions were obtained: a dark green and resinous *n*-hexane phase, a green and a less resinous ethyl acetate phase, and, finally, a yellow–brown colored hygroscopic aqueous phase. Afterwards, the aqueous phase and the ethyl acetate phase were subsequently purified by hydrophobic resin adsorption (XAD-16). The column was rinsed exhaustively with bi-distilled water and elution was executed with methanol and ethyl acetate. In general, we obtained three fractions by this purification step: an aqueous layer (215 g, 71%) and a purified methanol layer (73.5 g, 24%) after purification of the previous aqueous methanol phase and a purified ethyl acetate layer (14 g, 5%) which was less green and resinous compared to the previous ethyl acetate fraction from liquid-liquid partition process. LC-ESI-MS measurements revealed that the aqueous layer mainly consisted of polar compounds, whereas the methanol and the ethyl acetate phase were characterised mainly by glycosidic bound polyphenols. In addition to the methanol fraction, the ethyl acetate fraction contained polyphenol aglyca.

### 2.1. HPCCC Separation of the XAD-16 Purified Methanol Fraction

The HPCCC separations were intended to reduce further the complexity of the methanol fraction for characterization of compounds detected by the previous untargeted LC-ESI-MS screening (results from our own study, data not shown here). Prior to the actual separation, different biphasic solvent systems were evaluated by experimental determination of partition coefficients (*K_D_*) for tentatively identified dihydro-isocoumarin glycosides (data not shown). Due to the rather polar nature of these compounds, the system consisting of ethyl acetate-*n*-butanol-water (6:4:10, *v*/*v*/*v*) was found to be suitable as *K_D_*-values were in the correct range (1.0–2.5) [40]. A total of 89 fractions were obtained from this separation, which were examined in more detail by means of thin-layer chromatography (TLC) and visualized with the spray reagent of anisaldehyde and sulphuric acid. Based on these results, we decided to merge successive fractions. Subsequently, the resulting 45 fractions were characterized by LC-MS analysis. Figure 1 summarizes the results by comparing the corresponding CCC fractions and the retention times from the respective LC-MS measurements of the tentatively identified and detected compounds. Overall, this separation did not lead to pure fractions directly useable for structure elucidation. Therefore, we further combined these fractions into eight major fractions for further purification by preparative HPLC. Nevertheless, the XAD-16 purified methanol extract was separated using HPCCC and resulted in fractions of lower complexity.

### 2.2. HSCCC Separation of the Purified Ethyl Acetate Fraction

During the purification of the ethyl acetate fraction, the focus was on the dihydro-isocoumarin aglyca. Therefore, separation of the ethyl acetate partition was carried out with a preparative HSCCC system, as larger amounts of sample material can be fractionated with higher chromatographic resolution. Evaluation of a suitable biphasic system was done by determination of the partition coefficients of targeted tentatively identified dihydro-isocoumarin aglyca (data not shown). A so called HEMWat system consisting of *n*-hexane-ethyl acetate-methanol-water (4:6:4:6, *v*/*v/v*/*v*) gave best results for these *Hydrangea* constituents. A total of 165 fractions were recovered from the *elution*- and *extrusion*-modes. All fractions were analyzed by TLC and LC-MS measurements and finally combined into eight fractions (cf. Figure 2), whereby three (fracs. 5, 6, 7) contained relatively pure products which were directly used for NMR structure elucidation.

### 2.3. Dihydro-Isocoumarins

The application of the methods reported in 2.1 and 2.2 yielded in a total of 13 dihydrocoumarins (8 glycosides and 5 aglyca). In addition to the quantitatively dominating compounds hydrangenol-8-O-glc (**1**, 62 mg), thunberginol G-3′-O-glc (**2**, 12 mg), and phyllodulcin-8-O-glc (**3**, 21 mg), also some minor compounds hydrangenol-4′-O-glc (**4**, 2 mg), phyllodulcin-3′-O-glc (**5**, 3 mg), thunberginol C-8-O-glc (**6**, 1 mg), thunberginol E-6-O-glc (**7**, 5 mg), thunberginol D-6-O-glc (**8**, 2 mg), thunberginol C (**9**, 10 mg), thunberginol E (**10**, 4 mg), thunberginol G (**11**, 3 mg), hydrangenol (**12**, 46 mg), and phyllodulcin (**13**, 20 mg) were isolated (cf. 1D/2D-NMR assignments in Appendix A). In the performed LC-MS measurements all substances revealed similar MS/MS-fragmentation pattern in a spectrum obtained at high energy with common neutral loss cleavages of 43.990 [CO_2_] as exemplified for thunberginol C (**9**) in Figure 3. This neutral loss is related to the α-cleavage of the carbonyl group in the lactone-ring (C-ring) of the dihydro-isocoumarin aglycone. The glycosides showed a typical neutral loss of 162.053 [C_6_H_10_O_5_] from the cleavage of a hexoside moiety in the spectrum obtained at high energy. Another remarkable feature of the lactone ring is the remaining C-3 stereocenter. Therefore, dihydro-isocoumarins can be distinguished in *R*- and *S*- enantiomers. It was reported that dihydro-isocoumarins with a 4′-hydroxyl group such as hydrangenol show a tautomer-like behavior and can, therefore, be isolated from aqueous solutions in a ratio of approx. 1:1 [36]. Phyllodulcin with 4′-methoxyl group shows tautomer-like behavior only under certain conditions. The *R*-/*S*- ratio enantiomers is usually 5:1 [36]. Solely chiral dihydro-isocoumarin glycosides can be separated into their enantiomers by achiral reversed-phase chromatography [35,36,49].

All isolated compounds (cf. Figure 4) were previously described and reviewed by Cicek et al., with the exception of thunberginol E-6-O-glc (**7**), which was described by Liu et al. as florahydroside I and II [13,50]. In general, the lack of authentic reference substances and respective mass spectrometric data prompted us to recover target substances on preparative scale for generation of a LC-MS based analytical library for future analytical studies.

### 2.4. Dihydro-Stilbenic Acids

A compound with *m*/*z* at 435.1306 [M − H] was isolated from HPCCC (F7) and subsequent preparative HPLC. A similar MS/MS fragmentation pattern was observed compared to dihydro-isocoumarins with neutral loss cleavages of 162.0539 [C_6_H_10_O_5_], 43.9897 [CO_2_] and 18.0107 [H_2_O]. In contrast to dihydro-isocoumarins, this compound revealed not a pronounced UV-absorption at λ 320 nm. Additionally, a neutral loss cleavage of 106.0417 [C_7_H_7_O] was observed. These phenomena can be explained by the fact that in this compound compared to the structure of dihydro-isocoumarins the ‘lactone ring’ is present in an opened form, which was confirmed by evaluation of the 1D-/2D-NMR spectra (cf. signal assignments Appendix A). Thus, this substance was identified as cudrabibenzyl A (**14**, 12 mg, cf. Figure 5), already described in *H. macrophylla* ssp. *serrata* by Shin et al. [25] and data were in complete agreement with our NMR results.

Additionally, the appropriate aglycone was isolated from the ethyl acetate partition by HSCCC from fraction F5. The structure was confirmed by 1D-/2D-NMR experiments and identified as 2,4-dihydroxy-6-[2-(4-hydroxyphenyl)ethyl]benzoic acid (**15**, 1 mg, cf. Figure 5). Mass spectrometric analysis revealed a [M − H]^−^ signal a *m*/*z* 273.0785 and characteristic neutral loss cleavages of 106.0424 [C_7_H_7_O], 43.9900 [CO_2_] and 18.0106 [H_2_O] (cf. Figure 3).

### 2.5. Dihydrostilbenoids and Related Substance

In addition to dihydro-isocoumarins and dihydro-stilbenic acids, a dihydro-stilbenoid was isolated and structurally identified as lunularin-4′-O-glucoside (**16,** 2 mg, cf. Figure 6) (NMR cf. Appendix A). The compound was recovered from HPCCC fraction F7, purified by preparative HPLC and detected with *m*/*z* at 375.1454 [M − H]^−^. Dihydro-stilbenoids such as lunularin-4′-O-glucoside were so far unknown metabolites in *H. macrophylla* ssp. *serrata*. However, it was already detected and isolated from the skins of *Allium cepa* [51]. Our results from NMR experiments agreed with those from literature.

In addition to **16**, dihydro-resveratrol-3-O-glucoside (**17,** 1 mg, cf. Figure 6) was isolated and elucidated from the same HPCCC fraction with a subsequent preparative HPLC clean-up step. Results from 1D-/2D-NMR experiments (cf. Appendix A) were in accordance to lunularin-4′-O-glucoside. Mass spectrometric analysis revealed a signal with *m*/*z* at 391.1401 [M − H]^−^ and both compounds showed neutral loss cleavages of 162.0535 [C_6_H_10_O_5_] and 106.0469 [C_7_H_7_O] in the spectra obtained at high energy, explaining a hexoside moiety loss and the cleavage of a hydroxyphenyl group. Dihydro-resveratrol-3-O-glucoside was described before in various plants [52,53,54,55].

Compound **18** was obtained from HPCCC fraction F1 and detected with *m*/*z* at 457.1720 [M + HCOOH-H]^−^. 1D-/2D-NMR experiments identified this compound as 4-hydroxy-8-[4-glucopyranosyl] hydroxyphenyl]octane-2,6-dione (**18,** 1 mg, cf. Figure 6), which represents a linear triketide glycoside. Useful neutral loss cleavages with 162.0534 [C_6_H_10_O_5_], 58.0420 [C_3_H_6_O] and 18.0105 [H_2_O], confirming the structure proposal. Polyketides are formed by condensation reactions of malonyl-CoA units and are subsequently folded and transferred into aromatic systems [56]. Therefore, we assumed that the dihydro-stilbenoid metabolites from *Hydrangea* are potentially formed via this triketide pathway.

### 2.6. Cyanogenic Glycoside

Successive separation of the XAD-16 purified methanol extract by combination of HPCCC (F2) and preparative HPLC delivered compound **19** that exhibited a complex low energy ESI-MS/MS spectrum. The compound was identified by 1D/2D-NMR experiments as taxiphyllin (**19**, 6 mg) with an estimated purity >95%. Assignments for ^1^H, ^13^C, and 2D-NMR correlations were summarized in Appendix A, and were in good agreement with data of previously described cyanogenic glycosides [31]. The detected [M − H]^−^ signal at *m*/*z* 310.0924 [C_14_H_17_NO_7_] showed low intensity compared to other signals in the mass spectrum. By calculating the molecular formula from the exact mass, other mass signals were explained as *m*/*z* 932.2945 [3M − H]^−^–C_42_H_51_N_3_O_21_, *m*/*z* 621.1937 [2M − H]^−^-C_28_H_34_N_2_O_14_, *m*/*z* 490.1568 [2M-C_8_H_7_NO-H]^−^–C_20_H_29_NO_13_, *m*/*z* 359.1189 [2M − C_16_H_14_N_2_O_2_-H]^−^–C_12_H_24_O_12_ (cf. Figure 7). Taxiphyllin (cf. Figure 8) and other cyanogenic glycosides were already described for *H. macrophylla* ssp. *serrata*. [30,31,37], and were suspected for causing food intoxications after consumption of sweet tea preparations [19].

### 2.7. Secoiridoid Glycoside

A secoiridoid glycoside was isolated from HPCCC fraction F4 with combination of preparative HPLC and yielded an approximated purity of 50%. The compound was detected with [M − H]^−^ at *m*/*z* 359.1345 and identified by 1D-/2D-NMR experiments as deoxyloganic acid (**20,** 1 mg, cf Figure 8). The 1D-/2D NMR assignments were given in Appendix A and consistent with the results from literature [57,58]. The mass spectrum obtained at high energy revealed neutral loss cleavages of 162.0351 [C_6_H_10_O_5_], 43.9898 [CO_2_] and 18.0104 [H_2_O], corroborating the glucoside and carboxyl moieties (cf. Appendix A). Besides **20**, loganic acid (**21**) was detected with *m*/*z* at 375.1297 [M − H]^−^. However, the yield was too low for NMR structure elucidation experiments.

Secoiridoid glycosides such as loganin, secologanin, secologanic acid, and sweroside, as well as n-butyl derivatives were previously isolated from *H. macrophylla* and *H. macrophylla* ssp. *serrata* [23,29]. In addition to these relatively simple secorididoid glycosides, other derivatives linked to hydroxyphenyl polyketides were described [8,34]. Hydramacroside B (**22**) with *m*/*z* at 655.2258 [M − H]^−^ in HPCCC F5 and hydrangenoside A (**23**) with *m*/*z* at 619.2397 [M − H]^−^ were detected in HPCCC fraction F7 in lower amounts. These compounds show characteristic MS/MS fragment pattern, given in Appendix A.

### 2.8. Flavonols

In the first three consecutive HPCCC fractions mass signals for flavonoid glycosides with *m*/*z* at 771.2013 [M − H]^−^ (F1), 625.1424 [M − H]^−^ (F2) and 609.1466 [M − H]^−^ (F3) were detected. Based on characteristic MS/MS fragmentation pattern, we tentatively assigned these substances to the group of flavonol glycosides. Further purification by preparative HPLC and subsequent structure elucidation by 1D-/2D-NMR (cf. Appendix A), quercetin-3-O-[(*α*-rhamnopyranosyl)-(1→6)-O-*β*-sophoroside] (**24**, 1 mg, cf. Figure 9) (*m*/*z* 771.2013 [M − H]^−^), quercetin-3-O-*β*-sophoroside (**25**, 12 mg, cf. Figure 9) (*m*/*z* 625.1424) and kaempferol-3-O-*β*-sophoroside (**26**, 2 mg, cf. Figure 9) (*m*/*z* 609.1466 [M − H]^−^) were identified and consistent with literature data [18,25]. However, long-range correlations in the HMBC between the anomeric protons and the flavonol aglyca were neither detectable for **24** nor in **26**. Flavonol-3-O-glycosides show preferred homolytic cleavages, while glycosides at position 7-O display the usual heterolytic bond cleavages [59,60]. In our spectra, we could only detect the fragments from the homolytic cleavage, which leads to the assumption that these compounds are exclusively 3-O-glycosides.

These three structurally very similar substances were clearly separated by HPCCC with the biphasic solvent system consisting of ethyl acetate-*n*-butanol-water (6:4:10; *v*/*v*/*v*), which demonstrated the chromatographic resolution of all-liquid methodology for pre-fractionation and for preparative isolation of pure compounds. Flavonol glycosides **24**, **25** and **26** were previously described in *H. macrophylla* ssp. *serrata* [18,25].

## 3. Materials and Methods

### 3.1. Plant Material

For this experiment, the *Hydrangea macrophylla* ssp. *serrata* cultivar ‘Yae-no-amacha’ was purchased from Kötterheinrich Hortensienkulturen (Lengerich, Germany). These plants were cultivated from the cuttings stage in a cultivation room under controlled conditions (average temperature: 18–22 °C, humidity approx. 60%, lighting 16 h with P1-500-VIS LED^®^ (Future LED) with energy of 560 µmol/m^2^s) at Symrise AG (Holzminden, Germany). The cultivation was initially done in 5 L plant pots with Floragard TKS 2^®^ Instant plus (Floragard, Oldenburg, Germany) soil and later planted out in beds in the botanical garden of Symrise AG in Holzminden (Germany). Plants of *H. macrophylla* ssp. *serrata* ‘Yae no amacha’ were pruned and sample was collected in August 2021. 1.2 kg of leaves were manually separated from the stems and subsequently placed in a drying cabinet at 40 °C over 72 h.

### 3.2. Chemicals

Acetonitril, methanol, *n*-hexane, ethyl acetate, *n*-butanol, and LC-MS grade water used for this experiment were of appropriate quality and purchased from Honeywell (Seelze, Germany). Ultrapure water (resistivity ≥ 15 MΩ cm) was obtained from an Arium water purification system (Sartorius, Göttingen, Germany). Formic acid was purchased from Thermo Fisher (Geel, Belgium), deuterated dimethylsulfoxide from Deutero (Kastellaun, Germany), and Amberlite XAD-16 hydrophobic resin from Alfa Aesar (Kandeln, Germany).

### 3.3. Extraction

The dried and pre-crushed leaves were extracted twice with 10 L of methanol each time under constant stirring for at least 2 h at room temperature. The extraction solution was then filtered, combined, and evaporated under vacuum.

### 3.4. Purification of the Crude Extract

Subsequently, the concentrated methanolic crude extract was suspended in bidistilled water (1 L) and extracted successively with *n*-hexane (5 × 100 mL) and ethyl acetate (3 × 100 mL) until the organic phases became visibly clearer. The remaining aqueous phase was then passed through a column filled with Amberlite XAD-16 (~750 g, 300 × 70 mm). Adsorbed extract was washed with water, eluted successively with methanol using two column volumes (0.5 column volumes per hour) for each step, respectively. The obtained organic phases were concentrated under vacuum and subsequently lyophilized. The same procedure was also used for the ethyl acetate fraction.

### 3.5. Separation of XAD-16 Enriched Methanol Extract by Use of High-Performance Counter-Current Chromatography (HPCCC)

Partition coefficients values (*K_D_*) of potential target compounds in the group of dihydro-isocoumarin glycosides were determined for different biphasic solvent systems by LC-MS analysis. Therefore, the sample was given into the biphasic system and vigorously shaken by a Vortex apparatus (approx. 10 s). The phases were allowed to separate, aliquots of phases were taken and diluted, filtered through a 0.2 µm PTFE filter (Xtra PTFE-20/25, Macherey and Nagel, Düren, Germany) and transferred into HPLC vials. Samples were measured by LC-MS, and signal areas of selected single ion traces of target compounds were used for the calculation of the CCC-solvent specific metabolite partition coefficients.

Afterwards, the separations of the purified methanol XAD-16 fraction was performed with the solvent system consisting of ethyl acetate-*n*-butanol-water (6:4:10; *v*/*v*/*v*). Shortly before use, solvent components were mixed for equilibration in a 2 L separation funnel and phase layers allowed to separate (approx. 10 min). The phase layers were separated and degassed by ultrasonication.

The separation of the methanol XAD-16 extract was performed on a semi-preparative multilayer double bobbin HPCCC instrument (Dynamic Extractions, Ltd. Spectrum, HPCCC, Tredegar, UK) with 125 mL coil column volume and PTFE tubing (23.5 m × 2.6 mm i.d.). A double piston HPLC-pump (RT. Ing MP-2001, Potsdam, Germany) was used for solvent delivery, a thermostat (RC 6 CS Lauda R. Wohser, Lauda-Königshofen, Germany) for temperature control (30 °C) of the HPCCC rotary chamber, for detection at λ 210 nm an UV/Vis detector (Knauer Well-Chrom Spectro-Photometer K-2500, Berlin, Germany), and a fraction collector (Pharmacia LKB Super Frac, Brommer, Sweden).

The HPCCC system was loaded with the upper more organic stationary phase in the head-to-tail mode at 20 mL/min. The lower more aqueous phase was used as mobile phase with a flow rate of 4.0 mL/min at a rotation velocity of 1600 rpm. The HPCCC system reached the hydrodynamic equilibration after 20 min with a stationary values *S_f_* of 66% (residual stationary phase 83 mL).

For sample preparation, the XAD-16 methanolic extract of *Hydrangea* (0.5 g) was dissolved in 5 mL of stationary and mobile phase, each and filtered through a 1.0 µm glass fiber filter (Xtra GF-100/25, Macherey and Nagel, Düren, Germany). The sample solution was transferred by a medical syringe into a 5 mL sample loop. Injection occurred via a low-pressure sample injection valve (Rheodyne, Cotati, CA, USA) directly to the phase equilibrated HPCCC system. The experiment was performed in a so-called elution-extrusion mode. In the elution-mode, 72 fractions were collected in 2 min time intervals. The extrusion-mode was initiated by pumping the stationary (upper) phase, enhancing the flow rate to 8.0 mL/min and reducing the rotation velocity to 400 rpm. During extrusion-mode 26 fractions were collected in 1 min time intervals. For recovery of sufficient yields of target metabolites, the HPCCC separation was performed twice. Fraction evaluation was done by thin-layer chromatography (TLC) and LC-MS analysis. Based on the findings, the HPCCC runs were pooled into 8 major fractions for further clean-up procedures.

### 3.6. Separation of Enriched Ethyl Acetate Extract by Use of High-Speed Counter-Current Chromatography (HSCCC)

In the first step, partition coefficients (*K_D_*) of potential dihydro-isocoumarin aglyca were determined for different biphasic solvent systems as described in Section 3.5.

The separation of the purified ethyl acetate extract was performed with a solvent system consisting of *n*-hexane-ethyl acetate-methanol-water (4:6:4:6; *v*/*v/v*/*v*). Mobile and stationary phase preparation was done as mentioned in the previous chapter (cf. 3.5).

2 g of the ethyl acetate extract were dissolved in aliquot volumes 12.5 mL of stationary and mobile phase, each, and filtered through a 1.0 µm glass fiber filter (cf. 3.5). The biphasic sample solution was filled in a 25 mL sample loop then injected via a low-pressure sample injection valve (Rheodyne, Cotati, CA, USA) without phase equilibration to the HSCCC system. The separation was performed on a preparative triple multilayer coil HSCCC instrument (model CCC-1000, Pharma-Tech Research Corp., Baltimore, MD, USA). The total dimension of the three connected coils equipped with polytetrafluoroethlylene (PTFE) tubings was determined to 165 m × 2.6 mm i.d. and a total volume of 850 mL. The device was further equipped with a single HPLC-pump, UV/Vis-detector and a fraction collector (cf 3.5). The lower phase was used as mobile phase in a head-to-tail mode with a flow rate of 4.0 mL/min using a rotation velocity of 800 rpm. HSCCC separation was performed in the elution-extrusion mode. During the elution mode 158 fractions were collected in 3 min time intervals. Extrusion mode was initiated by pumping of stationary phase at a flow rate of 8.0 mL/ min and reduced rotation velocity of 200 rpm. In the extrusion mode, 48 fractions were collected in intervals of 1 min. Aliquots of every second fractions were evaluated by TLC and LC-MS analysis. Based on the results, the whole chromatographic run was combined to 8 major fractions.

### 3.7. Preparative HPLC

For further clean-up, HPCCC and HSCCC fractions were separated by preparative C18-reversed phase HPLC. Separations were carried out using HPLC system from Knauer (Berlin, Germany) consisting of a preparative HPLC pump (Knauer Pump P 2.1L, Knauer Gerätebau GmbH, Berlin, Germany), solvent degasser (Knauer), UV-detector (Knauer smartline UV-detector 2600), and fraction collector (Foxy R1, Teledyne Isco, Lincoln, NE, USA). The system was equipped with a preparative column (250 × 21.1 mm, 5 µm, 100 Å, Luna C18, Phenomenex, Aschaffenburg, Germany), with a pre-column of the same material with a flow rate of 15.0 mL/min at ambient temperature. As mobile phase water (eluent A) and acetonitrile (eluent B) with a linear gradient from 0.0 min–30% B, 30.0 min–40% B, 30.5 min–95% B, 40.0 min–95 % B was used with a pre-injection time of 5 min.

### 3.8. Structure Elucidation

#### 3.8.1. UPLC-ESI-IMS-QToF

Measurements were carried out using a Waters Acquity UPLC I-class system consisting of a binary pump, an autosampler, a column manager, and a PDA detector which is coupled to a Vion ESI-IMS-QToF mass spectrometer via a Z-spray electrospray ion source (Waters Corp., Milford, MA, USA). UNIFI software (version 1.9.13.9, waters, Milford, MA, USA) was used for instrument control, data acquisition, and data evaluation. The mass spectrometer operated in low mass range (100–1200 *m*/*z*), with the sensitivity analyzer mode and standard transmission mode. Instrument setup was performed using the ToF Instrument Service Sample Kit (Waters) before each analysis according to the manufacturer’s recommendations. The resolution at *m*/*z* 556 was determined to nearly 40,000 FWHM. Lockmass correction was performed automatically in 5 min intervals of through the reference sprayer with a solution of leucine enkephaline (54 nM in acetonitrile/ water, 50/50 (*v*/*v*) + 0.1% formic acid) at a flow rate of 10 µL/min.

Samples were analyzed with UPLC using following chromatographic conditions. Column: Phenomenex (Torrance, CA, USA) Kinetex RP-18 (100 × 2.1 mm, 1.7 µm, 100 Å; column temp.: 50 °C; inj. volume: 1 µL; mobile phase: A—water + 0.05% formic acid, B—acetonitrile + 0.05% formic acid; gradient mode: 0.00 min–0% B, 11.58 min–50% B, 11.70 min–95% B, 15.00 min–95% B; flow rate: 0.55 mL/min; and pre-inj. time: 3 min. Instrument settings for the mass spectrometer were: capillary voltage—3.00 kV (neg. mode), 2.20 kV (pos. mode), sample cone voltage: 40 V, source temp.: 120 °C, desolvation temp.: 500 °C, cone gas 50 L/h, desolvation gas 800 L/h, scan time 0.1 s, collision energy low energy: 6 V, and high energy ramp 20–50 V. MS/MS experiments were acquired data independently in HDMS^E^ mode.

#### 3.8.2. NMR

The samples for NMR spectroscopy were dissolved in *d_6_*-dimethylsulfoxide and transferred to a 5 mm sample tube. 1D/2D-NMR spectra including ^1^H-, ^13^C-, DEPT-135, ^1^H-^13^C-HSQC, ^1^H-^13^C-HMBC, ^1^H-^1^H-COSY were recorded at 25 °C on a Bruker 600/54 Ascend 4K spectrometer (Bruker BioSpin GmbH, Rheinstetten, Germany) operating at B_0_ = 14.4 T. The resonance frequencies correspond to 600 MHz for ^1^H and 151 MHz for ^13^C, respectively. A Prodigy cryoprobe was used to obtain a better signal-to-noise ratio. The spectra were referenced to solvent signals δ_H_ = 2.50 ppm and δ_C_ = 39.52 ppm for dimethylsulfoxide.

## 4. Conclusions

The two-dimensional purification workflow combining HPCCC/HSCCC and preparative HPLC, led to the isolation and structural characterization of 23 secondary metabolites in the methanol-based extract of *Hydrangea macrophylla* ssp. *serrata* comprising six natural products classes. In our previous study, untargeted UPLC-ESI-IMS-QToF analysis retrieved a list of compounds that differ significantly between the investigated cultivars. The tentatively identified compounds and some unknown compounds were structurally characterised in this work by 1D/2D-NMR and UPLC-ESI-IMS-QToF experiments. In addition, UPLC-ESI-IMS-MS/MS data were presented here for a variety of dihydro-isocoumarins (retention time values and ion-mobility collision-cross-sectional values—CCS values, cf. Appendix A), which could be used for identification in further studies. Furthermore, isolated compounds from the group of dihydro-stilbenoids and a triketide glycoside that were described in *Hydrangea macrophylla* ssp. *serrata* for the first time.

HPCCC separations were successfully applied as an efficient pre-fractionation method in the two-step isolation approaches. The biphasic solvent system was particularly powerful for the separation of three structurally very similar flavonol-3-O-glycosides.

HSCCC was used for the targeted isolation of dihydro-isocoumarin aglyca from the ethyl acetate partition of the methanol extract. Thunberginol C and thunberginol E were isolated in almost pure status for further in-depth structural characterization. Based on this proceeding, further optimization for the preparative isolation of specific dihydro-isocoumarin aglyca could be developed. Reference substances especially for dihydro-isocoumarins are of interest, which are difficult to access via a synthetic route.

## Figures and Tables

**Figure 1 molecules-27-03424-f001:**
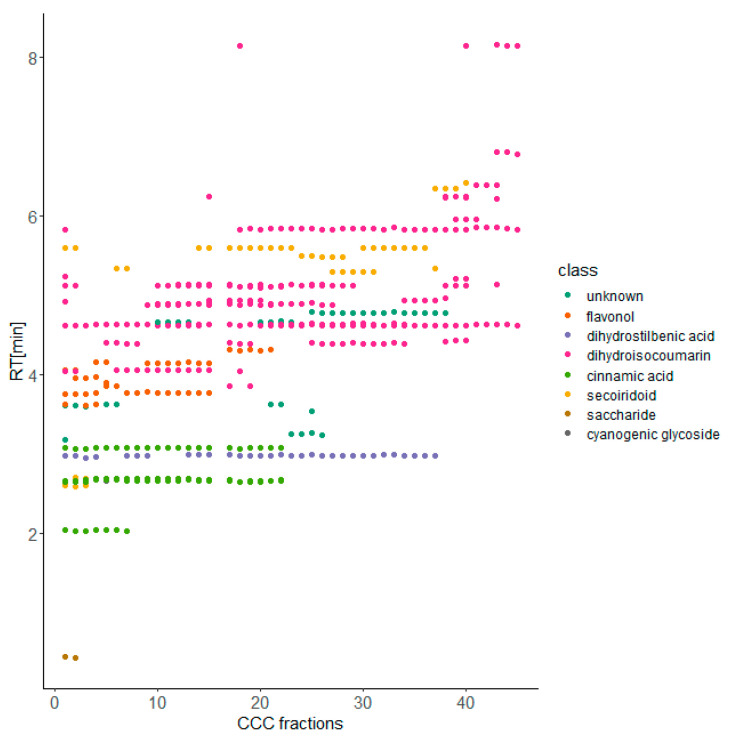
2D-chromatography plot of selected natural product classes in HPCCC-fractions (*x*-axis) versus LC-MS retention times (*y*-axis) (*Hydrangea* methanol extract XAD-16).

**Figure 2 molecules-27-03424-f002:**
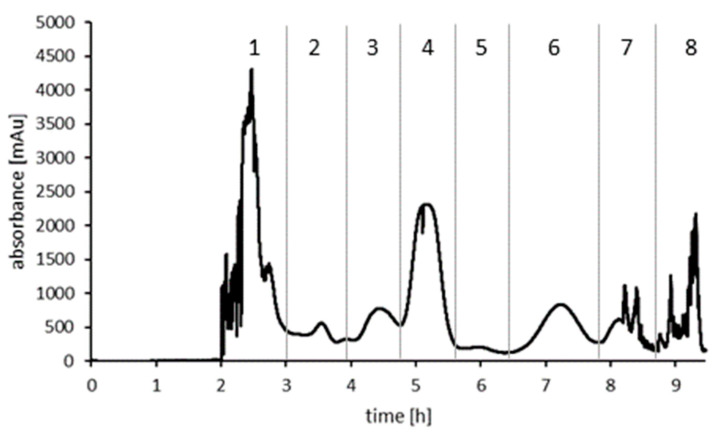
HSCCC chromatogram of the ethyl acetate fraction. Vertical lines represent combined fractions.

**Figure 3 molecules-27-03424-f003:**
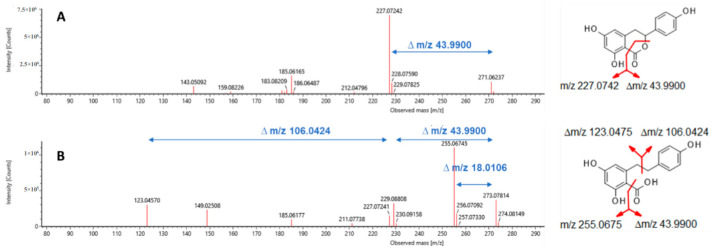
ESI-HDMS^E^ fragmentation of (**A**) thunberginol C (**9**), and (**B**) 2,4-dihydroxy-6-[2-(4-hydroxyphenyl)ethyl]benzoic acid (**15**).

**Figure 4 molecules-27-03424-f004:**
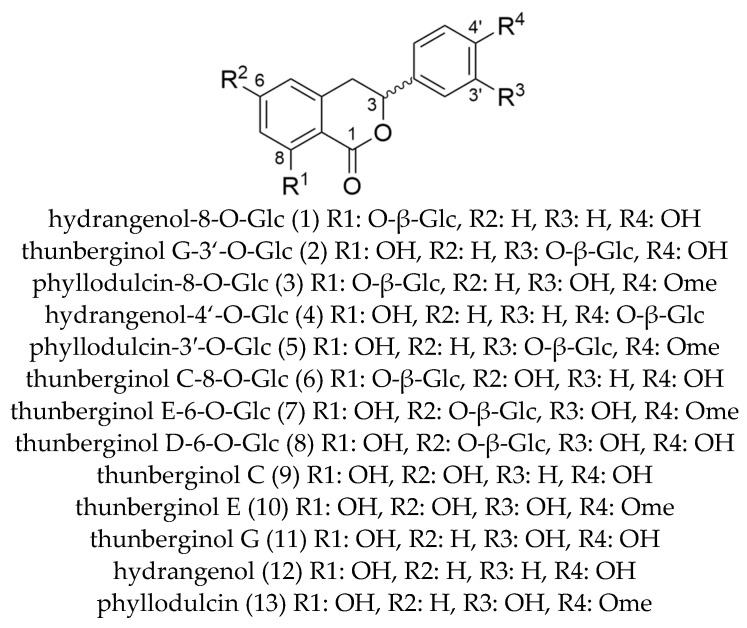
Molecular structures of dihydro-isocoumarins isolated from *Hydrangea macrophylla* ssp. *serrata*.

**Figure 5 molecules-27-03424-f005:**
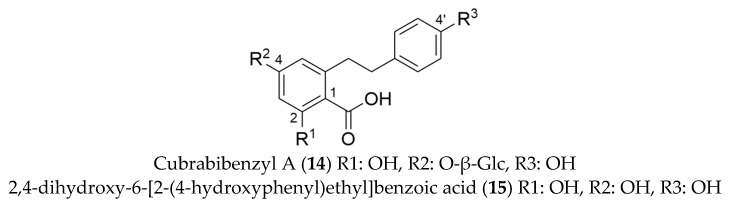
Molecular structures of dihydro-stilbenic acids isolated from *Hydrangea macrophylla* ssp. *serrata*.

**Figure 6 molecules-27-03424-f006:**
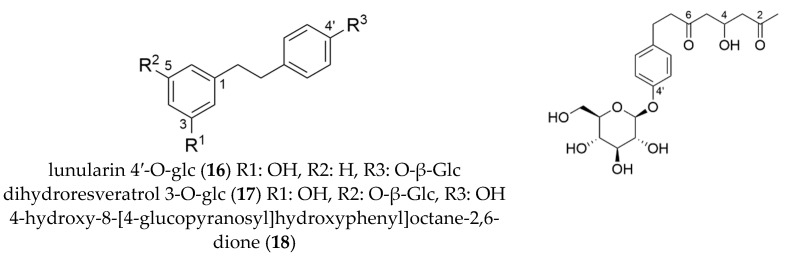
Molecular structures of dihydrostilbenoids and a related substance isolated from *Hydrangea macrophylla* ssp. *serrata*.

**Figure 7 molecules-27-03424-f007:**
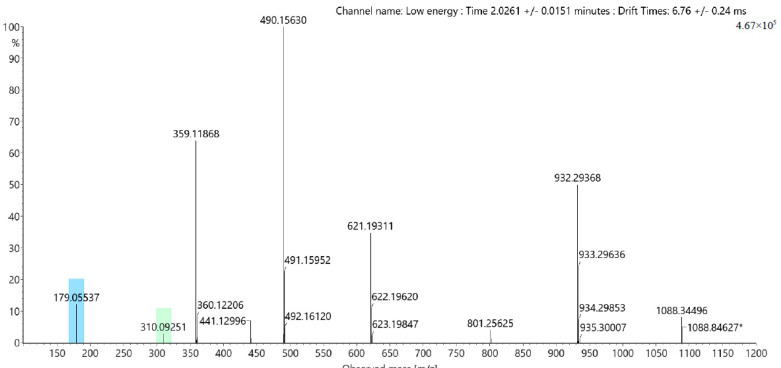
Low energy ESI MS/MS spectrum of taxiphyllin (**19**).

**Figure 8 molecules-27-03424-f008:**
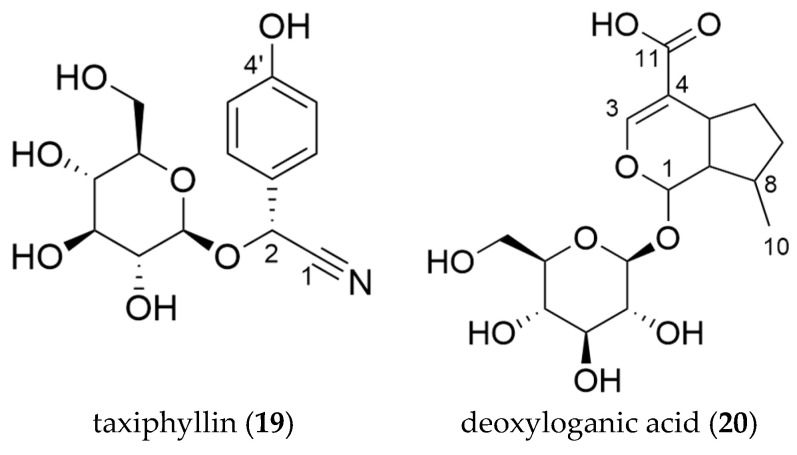
Molecular structures of taxiphyllin and deoxyloganic acid isolated from *Hydrangea macrophylla* ssp. *serrata*.

**Figure 9 molecules-27-03424-f009:**
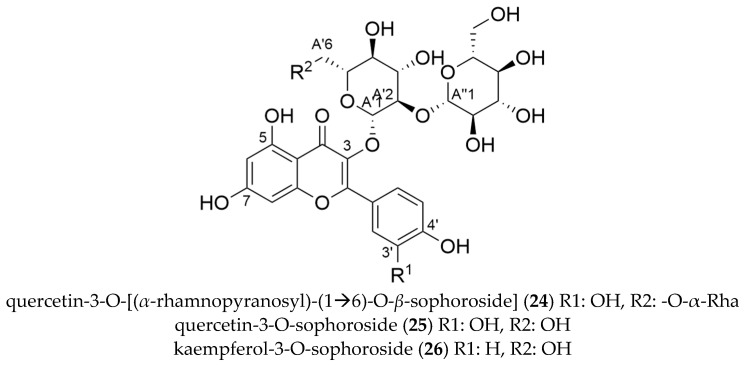
Molecular structures of flavonol glycosides isolated from *Hydrangea macrophylla* ssp. *serrata*.

## Data Availability

The data presented in this study are available in Appendix A.

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
