# Peer review of "Separation of Dihydro-Isocoumarins and Dihydro-Stilbenoids from Hydrangea macrophylla ssp. serrata by Use of Counter-Current Chromatography"

_molecules, 2022, doi:10.3390/molecules27113424_

Round 1

Reviewer 1 Report

The research provides new ideas and data for further related studies. But the discussion can be improved.

Line 97. Would methanol extraction result in a low extracting rate of polar compounds? Or say, would it be better to extract using methanol-water solvent? Also, Line 91-115 read like 'material and method' instead of 'result and discussion.

Line 116. section 2.1. Although the results have been presented, it is a lack of citations and comparison with other published references for a more professional, comprehensive, and better discussion. 

Line 153. Figure 2 can be improved to avoid a washout look. The border should be deleted, the data line should be black, a gray line can be deleted, a black y-axis trick should be added, etc.

Line 188. It would be better to provide a table listing all critical mass fragments in the supplementary materials.

Line 269. should be a new paragraph.

Line 333. Please provide the height and diameter of the resin column and the loading, eluting speed in resin bed volume (BV) per hour.

Line 460. using first-person is fine, but it would be better not too much for academic writing.

Reviewer 2 Report

In the presented manuscript, the authors clearly and in detail described the process of purification and isolation of the presented dihydro-isocoumarins and dihydro-stilbenoids using different chromatographic techniques. The presented results can be of great benefit to other researchers dealing with the chemistry of secondary metabolites and their purification processes.

There are a couple of typographical errors in the manuscript that need to be corrected. For example, in line 454. it says 1H-,13C-HSQC, 1H-,13C-HMBC and it should 1H-13C-HSQC, 1H-13C-HMBC.

I suggest authors to specify the isolated amounts of the individual components and provide NMR spectral for the newly isolated.

I highly recommend this manuscript for publication.

Reviewer 3 Report

The manuscript presents a continuation of previous studies in terms to determine dihydro-isocoumarins and dihydro-stilbenoids from plant samples. The interesting part is of course using CCC to identify metabolites using HP and HSCCC together with NMR and LC-MS. In general, manuscript is well planned and organized, I have only some minor comments listed below:

- I would suggest using past simple tense, instead of present perfect in some places, like line 123 – has been found -> was found

- high energy spectrum -> spectrum obtained at high energy

- l 192 similar fragmentation patter was observed on mass spectrum

Round 2

Reviewer 1 Report

The manuscript has been greatly improved, and I can accept it in its current form.